# Nuclear Transport of Respiratory Syncytial Virus Matrix Protein Is Regulated by Dual Phosphorylation Sites

**DOI:** 10.3390/ijms23147976

**Published:** 2022-07-19

**Authors:** Reena Ghildyal, Michael N. Teng, Kim C. Tran, John Mills, Marco G. Casarotto, Philip G. Bardin, David A. Jans

**Affiliations:** 1Centre for Research in Therapeutic Solutions, Faculty of Science and Technology, University of Canberra, Canberra 2617, Australia; 2Department of Internal Medicine, Morsani College of Medicine, University of South Florida, Tampa, FL 33612, USA; mteng@usf.edu (M.N.T.); kteng@health.usf.edu (K.C.T.); 3Faculty of Medicine, Monash University, Burnet Institute for Medical Research, The Alfred Hospital Department of Infectious Diseases, Melbourne 3004, Australia; john.mills@honorary.burnet.edu.au; 4Research School of Biology, Australian National University, Canberra 2601, Australia; marco.casarotto@anu.edu.au; 5Monash Lung & Sleep and Hudson Institute, Monash University, Melbourne 3181, Australia; philip.bardin@monashhealth.org; 6Nuclear Signalling Lab., Department of Biochemistry and Molecular Biology, Monash University, Melbourne 3181, Australia; david.jans@monash.edu

**Keywords:** RSV matrix protein, CK2 phosphorylation, nucleocytoplasmic shuttling

## Abstract

Respiratory syncytial virus (RSV) is a major cause of respiratory infections in infants and the elderly. Although the RSV matrix (M) protein has key roles in the nucleus early in infection, and in the cytoplasm later, the molecular basis of switching between the nuclear and cytoplasmic compartments is not known. Here, we show that protein kinase CK2 can regulate M nucleocytoplasmic distribution, whereby inhibition of CK2 using the specific inhibitor 4,5,6,7-tetrabromobenzo-triazole (TBB) increases M nuclear accumulation in infected cells as well as when ectopically expressed in transfected cells. We use truncation/mutagenic analysis for the first time to show that serine (S) 95 and threonine (T) 205 are key CK2 sites that regulate M nuclear localization. Dual alanine (A)-substitution to prevent phosphorylation abolished TBB- enhancement of nuclear accumulation, while aspartic acid (D) substitution to mimic phosphorylation at S95 increased nuclear accumulation. D95 also induced cytoplasmic aggregate formation, implying that a negative charge at S95 may modulate M oligomerization. A95/205 substitution in recombinant RSV resulted in reduced virus production compared with wild type, with D95/205 substitution resulting in an even greater level of attenuation. Our data support a model where unphosphorylated M is imported into the nucleus, followed by phosphorylation of T205 and S95 later in infection to facilitate nuclear export and cytoplasmic retention of M, respectively, as well as oligomerization/virus budding. In the absence of widely available, efficacious treatments to protect against RSV, the results raise the possibility of antiviral strategies targeted at CK2.

## 1. Introduction

Respiratory syncytial virus (RSV) is a common cause of lower respiratory tract disease in infants [1], as well as eliciting significant morbidity and mortality in the elderly and immunocompromised adults [2]. RSV is an enveloped virus that belongs to the Orthopneumovirus genus of the *Pneumoviridae* family. Its nonsegmented negative-sense RNA genome [1] encodes nine structural proteins, comprising the envelope glycoproteins (F, G, and SH), the nucleocapsid proteins (N, P, and L), the nucleocapsid-associated proteins (M2-1 and M2-2), and the matrix (M) protein and two nonstructural proteins (NS-1 and NS-). M protein is a major structural protein within the virion and has a key role in RSV assembly, through interactions with the envelope glycoproteins, nucleocapsid, and the host cell membrane and cytoskeleton [3,4,5,6,7,8,9,10,11,12].

Previously, we showed that M protein localization in the nucleus at the early stages of infection correlates with decreased host cell transcription, correlating strongly with M association with the host chromatin [13,14,15]. Interestingly, targets of M’s transcriptional repressive action in this context include nuclear-encoded mitochondrial gene products, which parallels strongly impaired mitochondrial function early in RSV infection [16,17,18]. Later in infection, M localizes largely to the cytoplasm, associating with nucleocapsid-containing cytoplasmic inclusions [6,8,13,19]. We have also shown that RSV M utilizes the conventional nuclear import pathway dependent on the nuclear import receptor importin β1 (IMPβ1) and the guanine nucleotide-binding protein Ran [19] for localization to the nucleus and exits the nucleus through a CRM1-/nuclear export sequence (NES)-dependent pathway [13]. Importantly, timely nucleocytoplasmic transport of RSV M is important for efficient and optimal RSV assembly, as indicated by reduced infectious virus production of recombinant RSV (rA2) with a nuclear import-deficient M and lack of viability of rA2 mutated in the M NES [13]. Regulation of switching between M nuclear import and export, the key to M’s roles in the nucleus and cytoplasm, is not understood. Phosphorylation is a well-characterized mechanism for modulating viral protein nuclear import [20,21,22,23], with protein kinase CK2 (CK2) implicated in a number of cases through consensus phosphorylation sites in close proximity to the nuclear targeting signals [24].

We previously reported that RSV M’s contribution to virus assembly/release is strongly dependent on the CK2 consensus site threonine 205 (T205), which may play a role in regulating M oligomerization [25]. Since T205 is located within the NES of RSV M [13], we postulated that phosphorylation at this site is likely to regulate the nucleocytoplasmic trafficking of the M protein, as is the case for several other viral proteins [25,26], including simian virus SV40 large tumor antigen (T-ag) and human cytomegalovirus processivity factor ppUL44 [20]. Here, we show for the first time that a consensus CK2 site serine at position 95 (S95), together with T205, can regulate the nucleocytoplasmic trafficking of M. Using quantitative imaging of cells transiently transfected to express GFP-fused M protein and mutant derivatives thereof, as well as cells infected with recombinant RSV/RSV mutant derivatives, we show that M subcellular localization can be regulated by these dual phosphorylation sites, the results supporting a model where T205 phosphorylation by CK2 may enhance export from the nucleus, followed by S95 phosphorylation to facilitate cytoplasmic retention, and subsequent oligomerization and virus budding.

## 2. Results

### 2.1. CK2 Activity Regulates RSV M Nuclear Localization

Building on previous observations [13,17,24], rA2-infected cells were treated with the CK2 specific inhibitor 4,5,6,7-tetrabromobenzo-triazole (TBB) for 12 h at early (6−18 h) or late (18−30 h) times p.i. and fixed and processed as samples for immunofluorescence to determine the effect of inhibiting CK2 on the subcellular localization of M protein. As used in this study, TBB does not lead to cell death [25,27]. As previously observed [6,15,19], M was predominantly nuclear at 6 h p.i. in untreated, infected cells (Figure 1a, column “No add”) but became progressively less nuclear, with characteristic cytoplasmic inclusions appearing from 18 h. After treatment with TBB at 6−18 h, M was perceptibly more nuclear at 18 h compared with untreated cells, even though a considerable portion of M remained associated with cytoplasmic inclusions (Figure 1a, column labelled TBB 6−18 h). This change in M subcellular localization was less obvious when cells were treated with TBB later in infection (Figure 1a, compare images for treatment with TBB 6–18 and 18–30 h). Quantitative analysis to determine the nuclear-to-cytoplasmic ratio (Fn/c) confirmed these observations (Figure 1b). In the absence of TBB, M was predominantly nuclear at 6 h p.i. (Fn/c of c. 2.7), and then became more cytoplasmic with time (Figure 1b, Fn/c of 1.7 and 0.6 at 18 and 30 h p.i., respectively). Treatment with TBB at 6−18 h p.i. (black columns) resulted in significantly (*p* < 0.0001) increased M nuclear accumulation (Fn/c of 3.5) compared with untreated cells, with this significantly (*p* < 0.001) higher accumulation compared with untreated cells still evident at 30 h p.i.

To confirm observations for M localization out of the context of RSV infection, Vero or A549 cells transiently transfected to express GFP-tagged M or GFP control proteins were treated without or with TBB and the localization of GFP-M was examined at 24 h post transfection (p.t.) by live cell confocal laser scanning microscopy, CLSM (Figure 1c,d). As expected, GFP alone was diffuse throughout the cell in the presence or absence of TBB, in stark contrast to the control molecule GFP-T-ag, which contains multiple CK2 sites known to facilitate nuclear accumulation [21,23,24], and accordingly showed reduced nuclear accumulation in the presence compared with the absence of TBB. GFP-M in contrast was mostly cytoplasmic in the absence of TBB (“No add”; Figure 1c) and became more nuclear when cells were treated with TBB. Image analysis confirmed these observations; no significant changes were observed in the absence or presence of TBB in the case of GFP (Fn/c of c. 1.0), in contrast to GFP-T-ag, which showed significantly (*p* < 0.0001) reduced nuclear accumulation in the presence compared with the absence of TBB (Figure 1d). Strikingly, TBB treatment resulted in a significant increase (*p* < 0.0003) in the nuclear levels of GFP-M in both Vero and A549 cells (Fn/c values of c. 1.4 and 0.8 in the presence compared with c. 0.7 and 0.5 in the absence of TBB treatment—see Figure 1d). The clear implication is that CK2 modulates nucleocytoplasmic trafficking of M independent of other RSV gene products in the infected cell. 

### 2.2. Dual CK2 Sites within RSV M Regulate Nucleocytoplasmic Distribution

Use of the ExPaSy public domain software enabled us to identify the serine at position 95 (S^95^LDE, single letter code) as a high-confidence CK2 site within RSV M, additionally to the previously reported threonine at position 205 (T^205^VTD). To begin to assess the functionality of this site, and its potential relationship to T205, truncation constructs (Figure 2a) with either one (GFP-M [1-183], GFP-M [110-256]) or both (GFP-M [1-256]) CK2 phosphorylation sites, or lacking both (GFP-M [110-183]), were expressed in Vero cells and their localization visualized using live-cell CLSM in the presence or absence of TBB (Figure 2b, images labelled “No add” or +TBB). All deletion constructs accumulated in the nucleus to levels higher than full length M (Fn/c values of 1 or higher compared with c. 0.45 in the absence of TBB; Figure 2b,c). GFP-M [1-183] and GFP-M [110-256], but not GFP-M [110-183], additionally showed significantly (*p* < 0.01) increased nuclear accumulation (Fn/c values of c. 1.4 and 1.2, respectively) in similar fashion to full-length GFP-M (Figure 2c). The results are consistent with the idea of two main CK2 phosphorylation sites in the M amino acid sequence (potentially S^95^LDE and T^205^VTD), within aa1-110 and 183-256, that modulate nucleocytoplasmic trafficking.

To test their roles in subcellular localization of M, S^95^ and T^205^ were substituted with nonphosphorylatable A residues or negatively charged D residues (to mimic phosphorylation at the respective sites), and the mutant derivatives were expressed as full-length GFP-fusions in transfected cells (Figure 3). A at either site (the S^95^A and T^205^A derivatives) had no significant effect on nuclear accumulation of M and did not appear to impair sensitivity to TBB, with both derivatives showing significantly increased (*p* < 0.0001) nuclear accumulation in the presence of TBB compared with its absence (Figure 3a,b).

The substitution of S^95^ by D (S^95^D) resulted in increased nuclear accumulation (Fn/c of c. 1.2) in the absence of TBB compared with wild type GFP-M (Fn/c of c. 0.5) and loss of responsiveness to TBB (Figure 3b). GFP-M:S^95^D additionally showed cytoplasmic aggregates resembling those of wild type M observed in a subpopulation of transfected cells (see Appendix A); these cells were not included in the Fn/c calculation. Substitution of T^205^ by D (T^205^D) also resulted in increased nuclear accumulation compared with GFP-M:T^205^A, but the levels of nuclear accumulation were not significantly higher than wild type M; it also had reduced sensitivity to TBB (Fn/c values of 0.58 and 0.74 in the absence and presence of TBB treatment).

The effect of the CK2 site mutations on M localization in cells were also assessed in the presence of the specific CRM1 inhibitor leptomycin B (LMB) (Figure 3a,c). As previously, LMB treatment resulted in significantly (*p* < 0.0001) increased nuclear accumulation of wild type GFP-M (Fn/c value of 1.1 compared with 0.5 in the presence and absence of LMB), with GFP-M:S ^95^A, -M:T ^205^A and -M:T ^205^D all showing similar responses. In stark contrast, GFP-M:S ^95^D was not sensitive to LMB treatment (Fn/c values of 1.27 and 1.21 in the presence and absence of LMB, respectively), implying that negative charge at S95, normally supplied by phosphorylation, inhibits M nuclear export via CRM1.

Overall, the lack of phosphorylation at either S^95^ or T^205^ (through A substitution) or negative charge at T ^205^ is not sufficient to alter the sensitivity to TBB of M subcellular localization. Negative charge at S ^95^, however, increases nuclear accumulation which is not sensitive to TBB or LMB treatment. This increased nuclear accumulation would appear to be in part due to the inhibition of M nuclear export and/or enhanced nuclear retention.

Analysis was extended to cells expressing GFP-M with dual S^95^ and T^205^ site substitutions; in all cases, combined mutation of both S^95^ and T^205^ eliminated TBB sensitivity (Figure 3d—compare the images labelled +TBB with those labelled “No add”; Figure 3e—compare clear and filled columns labelled AA, AD, DA, DD), consistent with the idea that CK2 regulation of M localization is exclusively through S^95^ and T^205^. While dual A (GFP-M:AA), AD (GFP-M:AD) or DD (GFP-M:DD) substitution had little effect on M localization compared with wild type M in the absence of TBB/LMB, DA substitution increased nuclear accumulation in similar fashion to substitution of D^95^ alone (Figure 3d,e); cytoplasmic aggregates were also observed in cells expressing GFP-M:DA but not GFP-M:DD. GFP-M:AA remained responsive to LMB, showing significantly (*p* < 0.05) increased nuclear accumulation, implying that nuclear export is completely comparable to that of wild type M (Figure 3d,f). In contrast, negative charge at T^205^, with nonphosphorylatable or phosphomimetic substitution at S^95^ (the AD and DD substitutions) abrogated responsiveness to LMB, indicating impaired CRM1-dependent nuclear export. M with negative charge at position 95 with a nonphosphorylatable mutant at position 205 (GFP-M:DA) retained LMB sensitivity.

Results overall are consistent with the idea that the S^95^ and T^205^ CK2 sites coregulate M nucleocytoplasmic trafficking, where negative charge mimicking phosphorylation at either S^95^ or T^205^ results in reduced M nuclear export, leading to increased nuclear accumulation.

### 2.3. RSV M S95 Is Essential for Virus Assembly

To assess the effects of the various M phosphorylation site substitutions on RSV infectious virus production, S^95^A and S^95^D mutations, with or without T^205^A, were introduced into the full-length genome of RSV rA2 in a reverse genetics system and multiple attempts were made to recover recombinant viruses. As for previous studies that were unable to recover rA2 carrying M:T^205^D [25], rA2 carrying M:S^95^D was not recovered, but the rA2-M:S^95^A and rA2-M:S^95^A/T^205^A) viruses were able to be recovered. Replication fitness was assessed in a multicycle replication assay in three different cell lines (Figure 4a), rA2-M:S^95^A showing very similar growth characteristics to rA2 in all. In contrast, rA2-M:S^95^A/T^205^A showed lower infectious virus production; this was quite pronounced in the A549 and HEp-2 lines, with a c. 100-fold reduced viral titer from day 2 (Figure 4a).

Since a role for T^205^ in the formation of higher order oligomers of M was implicated, we hypothesized that the introduction of a second phosphomimetic mutation at T^205^ could potentially compensate to some extent for the S^95^D mutation. This proved to be the case, in that, in contrast to rA2-M:S^95^D, rA2-M:S^95^D/ T^205^D could be recovered from Vero cells. Replication assays indicated significantly (*p* < 0.001) reduced fitness compared with rA2 in all three cell lines (Figure 4a). The inability to spread in the multicycle assay may have been due to reduced ability to bud from infected cells, or formation of a high proportion of defective infectious virus particles. To gain more insight into this, a budding assay was used, wherein HEp2 cells were infected with recombinant viruses at an MOI of 3, titer of released virus, and the level of.

RSV structural proteins N, P, and M were analyzed in whole-cell lysate and pelleted virus from culture supernatant (Figure 4c) by Western blotting. There was no difference in infectious virus titer between rA2 and rA2-M:S^95^A at 24 h p.i. (1−2 × 10^6^ pfu/mL in both cases), rA2-M:S^95^A/T^205^A, in contrast, had markedly reduced titer (0.6 × 10^6^ pfu/mL), while rA2-M:S^95^D/T^205^D was below the level of detection in the assay (<0.7 pfu/mL). N, P, and M were present in all samples of whole-cell lysates; N was also present in all samples of pelleted virus. P and M, in contrast, were detectable in all pelleted virus samples except that from cells infected with rA2-M:S^95^D/T^205^D. That there are equivalent levels of RSV structural proteins (N, P) in cells infected with rA2 and mutant viruses (Figure 4c, upper blot) indicates that the reduced titer of mutant viruses is not due to altered replication. The results clearly imply that M with negative charge at both S^95^ and T^205^ is not incorporated efficiently into the budding virus.

## 3. Discussion

This study establishes for the first time that S^95^ and T^205^ are key consensus CK2 phosphorylation sites in M that together regulate subcellular localization and thereby function in viral infection. Our data also suggest a role for phosphorylation at S^95^, along with that shown for T^205^ [25] in the oligomerization of M, with implications for RSV assembly. Our findings provide the mechanistic basis to understand previous work establishing that RSV M is transported into and out of the nucleus of infected cells early and later in infection, respectively, correlating with reduced host cell transcription early [15] and cytoplasmic RSV assembly [6] later in infection. The nuclear import and export of M had been previously established to occur through IMPβ1 [19] and CRM1 [13], respectively. Although the key importance of these pathways had been demonstrated by the fact that rA2 mutated in the NLS of M is attenuated, while rA2 with a nonfunctional NES is not viable [13], the mechanism of switching between nuclear import and export of RSV M had not been addressed. The present study shows for the first time that the S^95^ and T^205^ CK2 sites are the key, and that this switching mechanism is critically important to the RSV infectious cycle, as shown by the fact that rA2 carrying dual-site phosphorylation site mutations are attenuated (Figure 4); both sites are essential to RSV, needing to work in synergy for optimal virus production. Finally, the CK2-specific inhibitor TBB has been shown to inhibit RSV infection, demonstrating CK2′s importance to the RSV infectious cycle [25].

Figure 5 represents a model for how phosphorylation by CK2 of M S^95^ and T^205^ regulates M nuclear trafficking as well as its role in virus assembly. Since TBB-specific inhibition of CK2 increases M nuclear accumulation, we propose that newly translated M lacking S^95^/T^205^ phosphorylation is the form of M imported into the nucleus early in infection through IMPβ1 [19]. Later in infection, phosphorylation of M at T^205^ and S^95^ reduces/blocks nuclear import and enhances cytoplasmic retention, leading to increased oligomerization of M. That phosphorylation of T^205^ and S^95^ occurs late in RSV infection is supported by the observation that M associated with viral ribonucleoprotein complexes in the cytoplasmic inclusion bodies [6] runs with a different electrophoretic mobility than M localized in the nucleus (Appendix A). Given that previous work indicates that negative charge at T^205^ inhibits M oligomerization [25] underlying the formation of filamentous RSV [28], we postulate that dephosphorylation of M at T^205^ takes place prior to virus budding. That negative charge/phosphorylation at T^205^ inhibits M oligomerization is supported by the finding that GFP-M:DA, but not GFP-M:DD, formed cytoplasmic aggregates (Figure 3d,e). Our finding that rA2-M S^95^D/T^205^D was able to be rescued but was defective in virus budding also lends support to the idea that negative charge at T^205^ inhibits oligomerization and filament formation. Notably, negative charge at T^205^ does not impair the dimerization of M, and the deletion mutants of M that are unable to dimerize do not form aggregates in the cellular context [29], suggesting that a dimerization event precedes the formation of oligomers.

When put in the context of the three-dimensional structure of RSV M [30,31] (see Appendix A), T^205^ is present in a shallow depression on the surface of the protein surrounded by glutamate at position 184, alanine at position 188, and serine at position 220 [25] and is accessible for phosphorylation. In contrast, S^95^ is buried within the M structure close to the dimerization surface in a lipophilic environment, surrounded by leucine at position 31, tryptophan at position 36, and phenylalanine at position 88, making this site intriguing. Phosphorylation at this residue would lead to a pronounced conformational change; interestingly, S^95^ is located at the interface of the proposed dimer [30]. It is therefore highly likely that dimer formation is influenced by or influences the phosphorylation of S^95^, as seen by the increased aggregation of GFP-M:S^95^D (Appendix A). A population of cells expressing GFP-M also contains similar aggregates that resemble (but are not identical to) initial stages of virus filament formation. We propose that S^95^ is phosphorylated later in infection probably via a change in conformation brought about by an as yet unknown mechanism, thus ensuring that M oligomerization occurs in a timely fashion to promote rather than inhibit assembly and budding.

It is striking that CK2-mediated phosphorylation is a common regulatory mechanism for nucleocytoplasmic trafficking for many viral and also cellular proteins [20,21,32,33,34,35,36]; this is presumably at least in part due to CK2′s largely constitutive activity in most cells [37]. Previous work by our group has shown that TBB inhibition of CK2 can limit RSV virus production [25]; the work here further supports the idea that CK2 phosphorylation of specific viral proteins is a viable target for therapeutic development, a focus of future work in this laboratory.

## 4. Materials and Methods

### 4.1. Cell Culture and Transfection

Human alveolar epithelial A549 cells, human epithelial Hep2 cells, and African green monkey kidney Vero cells were obtained from European tissue culture collection (ECCAC) and used for rescue of recombinant virus, virus culture, and all infections. Cells were cultured in a humidified 5% CO_2_-atmosphere in Dulbecco-modified Eagle’s medium containing 10% fetal bovine serum supplemented with penicillin, streptomycin, and neomycin (Sigma-Aldrich Pty Ltd., Sydney, Australia).

Cells were transfected using Lipofectamine (Invitrogen, Melbourne Australia; 1:1 mix of DNA and reagent) 24 h before analysis by CLSM; see below. Where relevant, LMB (2.8 ng/mL) was added at 18 h post-transfection for 6 h, or TBB (25 μm) at 6 h post-transfection for 18 h, prior to CLSM and image analysis.

### 4.2. Virus Culture

RSV titers in cell lysates (cell-associated virus) or supernatant (released virus) were determined as described in [38,39]. Briefly, the culture supernatant was collected, cells washed, and lysed in serum-free medium containing SPGA (218 mM sucrose, 7.1 mM K2HPO4, 4.9 mM sodium glutamate, 1% (wt/vol) bovine serum albumin). Lysate and supernatant were clarified by centrifugation and stored at −80 °C until required. The infectious RSV titer was determined by triplicate immunoplaque assay on Vero cells, as previously in [25].

### 4.3. M Expression Constructs

The M coding sequence was generated by PCR from full-length rA2 cDNA [40] and introduced into the Gateway compatible entry vector pDONR207, followed by recombination into the pEPI-DESTC mammalian expression destination vector [19]. QuikChange site-directed mutagenesis (Promega) was used to perform site-specific mutations of the CK2 phosphorylation sites in the full-length M clone, as described in [13]. Each reaction contained 10 ng of template DNA and 80 ng of each primer. Resultant clones were confirmed by restriction analysis followed by DNA sequencing. The plasmid pEPI-Rev encoding the GFP-Rev(2–116) fusion protein with the CRM1-recognized NES in pEPI-DESTC [41] was used as a positive control for inhibition of nuclear export by LMB; due to its strong NES, Rev is largely cytoplasmic in the absence of LMB but strongly nuclear/nucleolar in its presence. Plasmid pEPI-Tag, encoding the GFP-SV40 T-ag (110–153) fusion protein containing the IMPα/β-recognized NLS was used as positive control for CK2 inhibition by TBB; strong nuclear localization of GFP-T-ag is reduced in the presence of TBB [42]. Plasmid pEGFP-C1 (Invitrogen) was used in all experiments as a negative control for LMB and TBB action. All expression constructs used in the study expressed their encoded proteins to similar levels as determined by Western analysis (data not shown).

### 4.4. Generation and Recovery of Mutant Recombinant RSV

Recombinant RSV strain rA2 has been described in [40]. The HindIII–NcoI cassette of the pEPI-M plasmids containing the S^95^ and/or T^205^ mutations were cloned into the shuttle vector pGEM-HX. The shuttle vectors so obtained were digested with SpeI and XhoI to subclone the coding sequences for the mutated M genes into vector D51 [43], digested using the same restriction sites. Finally, the AatII-XhoI fragment of the mutant D51 constructs was cloned into the full-length antigenome (D53). The mutant D53 plasmids were used for RSV recovery as described previously in [43,44].

Recombinant wild type (rA2) or the mutants (rA2-M:T^205^A, rA2-M:S^95^A, rA2-M:S^95^A/T^205^A, rA2-M:S^95^D/T^205^D) were used to infect cells at an MOI of 0.1. Virus was harvested at the times indicated, and titer was determined in immunoplaque assays [13].

### 4.5. Virus Budding Assay

Virus particles were collected from the culture supernatants from cells infected with rA2 or mutants at an MOI of 3 by pelleting through a 30% sucrose cushion, lysed in Laemmli buffer and used for Western analysis. Blots were blocked for 1 h in 4% skim milk (Diploma) in PBS, followed by incubation with anti-RSV antibody overnight at 4°C with rocking. Blots were incubated with species-specific secondary antibodies conjugated to horseradish peroxidase, followed by washing and detection of bound antibodies with Enhanced Chemiluminescence (ECL, Perkin Elmer). Bound protein images were captured on the Licor Odyssey Fc, and digital images were analyzed using ImageStudio software (Licor). Infectious virus titers were determined in the culture supernatant as above.

### 4.6. Infection and Immunofluorescence

Cells were grown to 80% confluence on glass coverslips before being infected at an MOI of 3 with rA2 [40]; certain cultures had 25 μM TBB added for 12 h at 6 or 18 h postinfection (p.i.). At the end of the 12 h incubation, the medium was replaced with fresh medium without TBB and cells were cultured up to 48 h p.i. At the indicated times, p.i., cells were fixed with 4% formaldehyde, permeabilized with 0.2% Triton X-100, and immunostained with an M-specific monoclonal antibody (MAbαM), a gift from Erling Norrby and Mariethe Ehnlund [45], and Alexa Fluor 488-conjugated secondary antibody (Invitrogen), followed by CLSM.

### 4.7. Confocal Laser Scanning Microscopy (CLSM) and Image Analysis

Fixed and live (transfected) cell samples were imaged as previously in [13], and ImageJ v1.62 public domain software was used as described previously in [39] to determine the nuclear/cytoplasmic fluorescence ratio (Fn/c), which was determined using the formula: Fn/c = (Fn-Fb)/(Fc-Fb), where Fn is the nuclear fluorescence, Fc is the cytoplasmic fluorescence, and Fb is the background fluorescence (autofluorescence).

### 4.8. Statistical Analysis

Graphpad Prism 6 was used for the determination of statistically significant differences using the Mann–Whitney test or two-way ANOVA as appropriate.

## Figures and Tables

**Figure 1 ijms-23-07976-f001:**
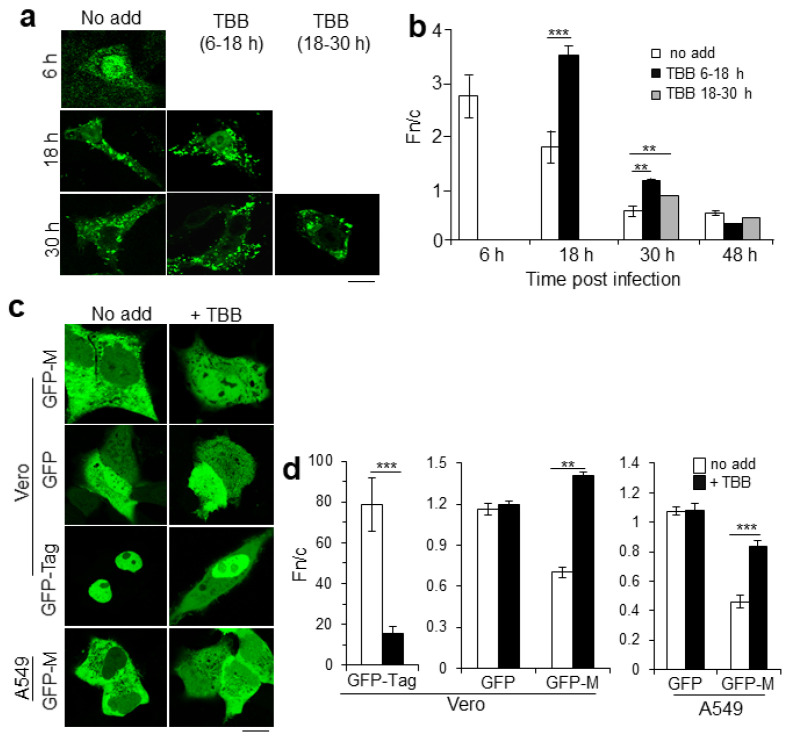
CK2 activity facilitates nuclear localization of RSV M. (**a**) Vero cells were infected with RSV (multiplicity of infection-MOI-of 3) as described in Materials and Methods, and after removal of virus inoculum, treated with the specific CK2 inhibitor TBB for 12 h either 6 (6−18 h, p.i.) or 18 (18−30 h.p.i.) h later. Cells were fixed at 6 h, 18 h, and 30 h p.i. and probed for localization of M by immunofluorescence followed by CLSM; representative images are shown. (**b**) Digital images such as those in (**a**) were analyzed as described in Materials and Methods for the nuclear to cytoplasmic ratio (Fn/c), and the histogram represents the mean ± SEM (*n* ≥ 30 cells). **, *p* < 0.001, ***, *p* < 0.0001. (**c**) Vero cells transiently transfected to express GFP alone, GFP-M, or GFP-T-Ag and A549 cells transfected to express GFP-M were treated without (no add) or with TBB as indicated prior to imaging by CLSM. Representative images are shown in (**c**), and quantitative analysis as per (**b**) is shown in (**d**) Scale bars = 10 μm.

**Figure 2 ijms-23-07976-f002:**
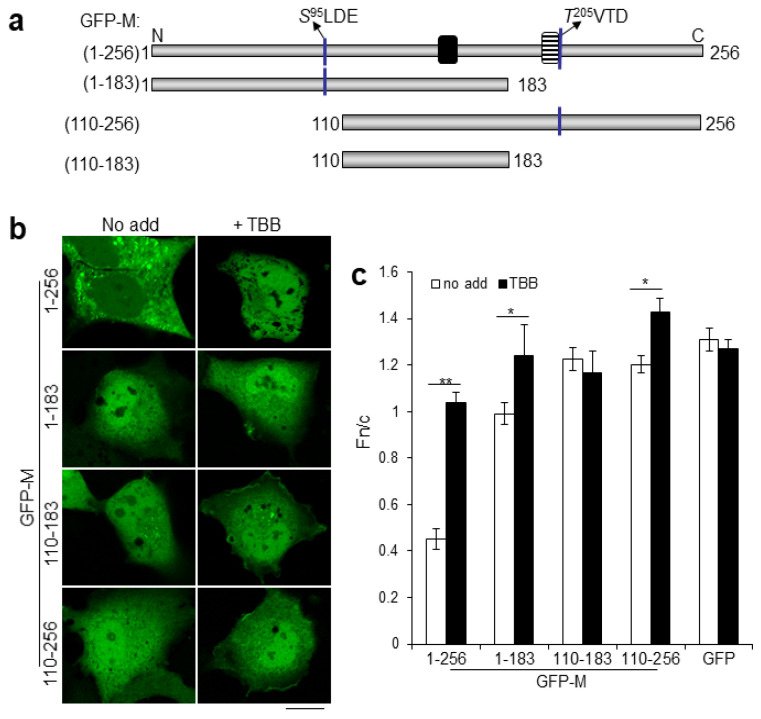
RSV M contains two key CK2 sites. (**a**) GFP-M constructs for mammalian cell expression used in this study lacking both or containing one or both of the key CK2 sites. (**b**) GFP fusion constructs from (**a**) were expressed in transfected Vero cells and localization of GFP fluorescence followed by live-cell CLSM 16 h later; representative images are shown. (**c**) Digital images such as those in (**b**) were analyzed as per the legend to Figure 1b. *, *p* < 0.01 **, *p* < 0.001. Scale bar = 10 μm.

**Figure 3 ijms-23-07976-f003:**
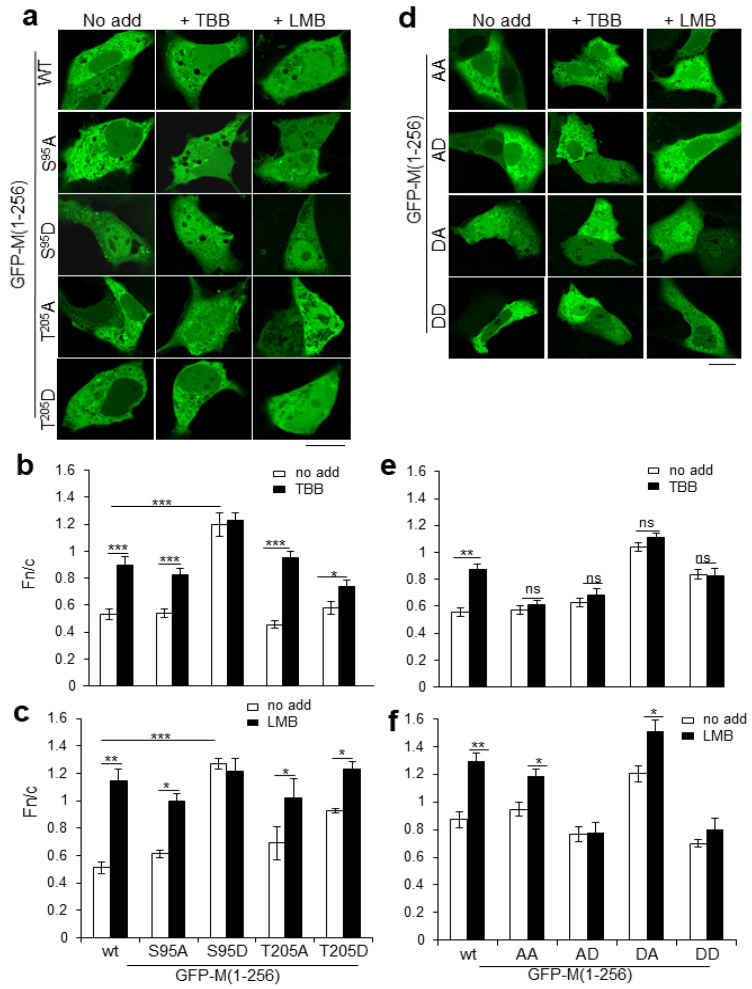
The S^95^ and T^205^ CK2 sites modulate RSV M nucleocytoplasmic distribution. (**a**,**c**) GFP fusion constructs of M (wt) and single-site mutated derivatives S ^95^ and T ^205^ were expressed in Vero cells and localization of GFP fluorescence followed by live-cell CLSM with and without TBB or LMB; representative images are shown in (**a**) and quantitative analysis as per the legend in Figure 1b in (**b**,**c**). * *p* < 0.05, **, *p* < 0.001, ***, *p* < 0.0001. S95A-Serine at position 95 mutated to Alanine, S95D-Serine at position 95 mutated to Aspartate, T205A-Threonine at position 205 mutated to Alanine, and T205D-Threonine at position 205 mutated to Aspartate. (**d**,**f**) GFP fusion constructs of double mutant derivatives of M mutated in both S^95^ and T^205^ were analyzed as per (**a**,**c**); representative images are shown in (**d**) and quantitative analysis in (**e**,**f**). * *p* < 0.05, **, *p* < 0.001, ns—not significant. AA-S95A/T205A, AD-S95A/T205D, DA-S95D/T205A, DD-S95D/T205D. Scale bars = 10 μm.

**Figure 4 ijms-23-07976-f004:**
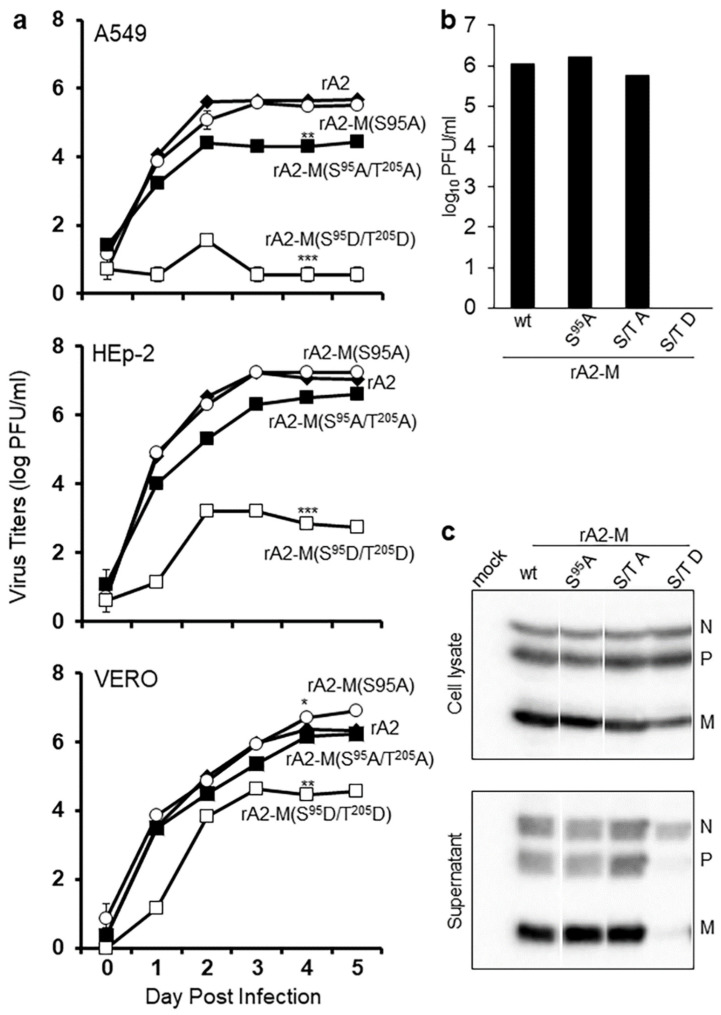
T^205^ and S^95^ substitutions attenuate RSV. (**a**) Wild type rA2 without or with S^95^ and/or T^205^ substituted by a nonphosphorylatable alanine (A) or with S^95^ and T^205^ substituted by the phosphomimetic aspartic acid (D), were rescued, used to infect Vero, A549 or HEp2 cells at an MOI of 0.1, supernatants collected at 24 h intervals for 5 days, and the plaque-forming units (pfu)/mL determined by immunoplaque assay on Vero cells. * *p* < 0.05, ** *p* < 0.001, *** *p* < 0.0001. (**b**) A549 cells were infected with the indicated viruses at an MOI of 3, culture supernatants were collected for immunoplaque assays. (**c**) Supernatants were also clarified, and virus particles pelleted through 30% sucrose (Supernatant). Cells were lysed with RIPA buffer (cell lysate). Cell lysates and pelleted particles were analyzed for N, P, and M by Western blotting. S95A-rA2 with M substituted by alanine at S95. S/T A-rA2 with S95 and T205 within M were both substituted by alanine. S/T D-rA2 with S95 and T205 within M were both substituted by aspartic acid.

**Figure 5 ijms-23-07976-f005:**
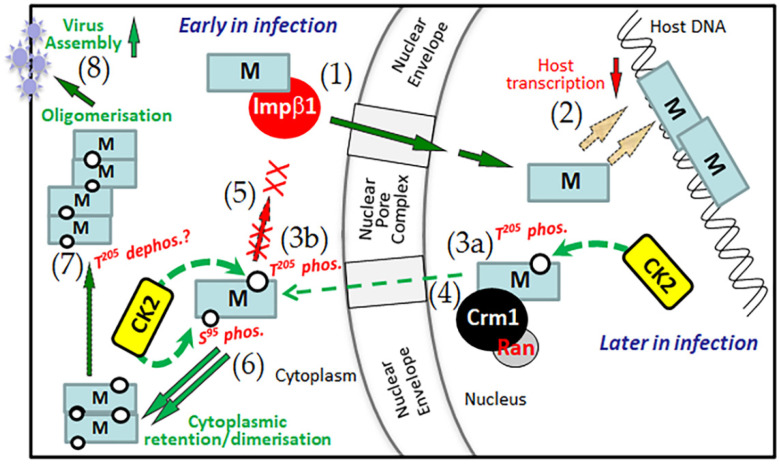
CK2 regulation of M nuclear trafficking and its role in virus assembly. Early in infection, newly translated M lacking phosphorylation at S^95^/T^205^ is imported into the nucleus through IMPβ1 (**1**) to associate with the host chromatin and affect host gene transcriptional inhibition (**2**). Later in infection, phosphorylation of M at T^205^ (dashed green arrows) serves as a switch, both in the nucleus (**3a**) where it enhances nuclear export through CRM1 (**4**) and in the cytoplasm (**3b**) where it inhibits nuclear import (**5**). Phosphorylation at S^95^ in the cytoplasm (**6**) facilitates cytoplasmic retention/dimerization, which favors M’s role as an adaptor in virus assembly. Although not completely clear, dephosphorylation of M at T^205^ by an unknown mechanism (**7**) facilitates M dimerization/oligomerization, which underlies the formation of filamentous RSV, leading to increased virus assembly (**8**). Inhibition of CK2 or prevention of phosphorylation at either site throughout the infectious cycle through mutation (both not shown) increase M nuclear accumulation, ultimately leading to reduced RSV virus production.

## Data Availability

All data are contained within the manuscript or Appendix A.

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
