# Peer review of "Nuclear Transport of Respiratory Syncytial Virus Matrix Protein Is Regulated by Dual Phosphorylation Sites"

_ijms, 2022, doi:10.3390/ijms23147976_

Round 1

Reviewer 1 Report

The authors of the manuscript ‘Nuclear Transport of Respiratory Syncytial Virus Matrix Protein is Regulated by Dual Phosphorylation Sites’ characterize the role of tow phosphorylation sites on the RSV M protein which is associated with the transport of viral genomic material from the nucleus to the extracellular environment. In a serious of experiments conducted in relevant cell lines, the authors demonstrate the importance of the amino acids Serine and Threonine as phosphorylation site on the M protein.

I have the following comments on the manuscript:

Figure 1:

1.       How do we know that when the authors inhibited protein M translocation, the aggregated protein did not modulate cells death?

2.       Does the authors have additional nucleic stain to clarify which cells are alive?

3.       In all the figures we do not know what the X of magnification is.

Figure 2:

1.       Assuming that S and T has a role in phosphorylation in would be helpful to add additional 3D picture of the protein and show the reader where the location of those amino acids is.

2.       When the authors show reduced transfection, do we know it is because of lack of infection? Enhanced cell death? The presented results do not answer this question.

3.       Did the authors consider measure the quantity of RSV transcripts as a measurement of viral replication? This would assist us to know if the changes in protein M effect budding, replication or both.

Figure S3:

1.       It is clear that the authors want to show a 3D picture of the protein, however, it is not clear what is the exact location of the mentioned AA, surface? Pocket? What is the surrounding environment?

Author Response

REVIEWER 1

 We thank the Reviewer.

Figure 1:

  1. The treatments with TBB have been previously shown not to be toxic – see Fig. 1C Bajorek et al J Virol 2014 (see also Tapia et al, PNAS, 2006). We now add this important point to the manuscript (with citations, page 2, line 91) and thank the reviewer for pointing out this omission.
  2. As per point 1, TBB is not toxic.
  3. We have now added scale bars to all figures and amended the legends accordingly. The omission of the scale bar in the original submission is regretted.

Figure 2:

  1. A 3D image of the M dimer showing Serine 95 and Threonine 205 was provided in the Supplementary Data (Figure S3) of the original submission. The Reviewer’s specific question relating to Figure S3 is addressed below.
  2. The Reviewer’s question about transfection is unclear - transfection levels are comparable in all cases (eg. see Figure 2B) - there is no infection in Figure 2. If the Reviewer’s question relates to Figure 4 (?), he/she can rest assured that the reduced replication of mutant viruses is not due to altered replication as is clear from the equivalent levels of RSV structural proteins (N, P) in wild type and mutant viruses (Figure 4C, upper blot). We have added this point to the manuscript (page 10, lines 259 – 261) and thank the Reviewer for bringing it to our attention.
  3. We did not quantify the RSV transcripts in infected cells. We do not expect that the change in M protein would impact replication as comparable levels of structural proteins were expressed in cells infected with the wild type or mutant proteins. In addition, we have previously shown (Bajorek et al, 2014, Journal of Virology) that mutation of threonine 205 to alanine does not lead to any change in replication.

Figure S3:

  1. The 3D structure shows the M dimer as a ‘mesh’ with serine at position 95 and threonine at position 205 shown as red and blue spheres respectively. As shown in the figure, threonine 205 is present on the surface in a pocket surrounded by Glutamate 184, Alanine 188 and Serine 220 (also see Bajorek et al 2014). Serine 95 is present buried inside close to the dimerization surface in a lipophilic environment, surrounded by Leucine 31, Tryptophan 36 and Phenylalanine 88. We have now added this information to the manuscript (page 11, lines 315 – 319) and thank the Reviewer for highlighting this point.

Reviewer 2 Report

The manuscript describes interesting results on the involvement of certain phosphorylation sites on transport between the nucleus and cytosol during RSV infection.

The manuscript is well written, but deserves modification based on my following comments.

Prefer the passive voice.

Could the authors discuss the implication and generalization of their results, between different skeletons?

Author Response

REVIEWER 2

We thank the Reviewer.

  1. We have changed some of the text to passive voice where we felt that it was appropriate to do so and did not result in an overly complex sentence construction. E.g page 2, line 87; page 8, line 223; page 10, line 249; page 11, line 332.
  2. In terms of discussing the implications of our study, we have expanded our description of Figure S3 as requested by Reviewer 1 - we think it makes clearer the structural aspects/role within the M dimer. The last paragraph of the Discussion covers the broader implications for the future, as requested.